# Role of Lipocalin-2 in Amyloid-Beta Oligomer-Induced Mouse Model of Alzheimer’s Disease

**DOI:** 10.3390/antiox10111657

**Published:** 2021-10-21

**Authors:** Heeyoung Kang, Hyun Joo Shin, Hyeong Seok An, Zhen Jin, Jong Youl Lee, Jaewoong Lee, Kyung Eun Kim, Eun Ae Jeong, Kyu Yeong Choi, Catriona McLean, Kun Ho Lee, Soo Kyoung Kim, Hae Ryong Lee, Gu Seob Roh

**Affiliations:** 1Department of Neurology, College of Medicine, Gyeongsang National University Hospital, Gyeongsang National University, Jinju 52727, Korea; 2Bio Anti-Aging Medical Research Center, Department of Anatomy and Convergence Medical Science, Institute of Health Sciences, College of Medicine, Gyeongsang National University, Jinju 52727, Korea; k4900@hanmail.net (H.J.S.); gudtjr5287@hanmail.net (H.S.A.); jyv7874v@naver.com (J.Y.L.); woongs1111@gmail.com (J.L.); kke-jws@hanmail.net (K.E.K.); jeasky44@naver.com (E.A.J.); 3University of Tennessee Health Science Center, Department of Anatomy and Neurobiology, Memphis, TN 38163, USA; zkim777@gmail.com; 4Gwangju Alzheimer’s & Related Dementia Cohort Research Center, Chosun University, Gwangju 61452, Korea; khaser@gmail.com (K.Y.C.); leekho@chosun.ac.kr (K.H.L.); 5Department of Pathology, The Alfred Hospital, Melbourne 3004, Australia; C.McLean@alfred.org.au; 6Department of Biomedical Science, Chosun University, Gwangju 61452, Korea; 7Korea Brain Research Institute, Daegu 41062, Korea; 8Department of Internal Medicine, College of Medicine, Gyeongsang National University Hospital, Gyeongsang National University, Jinju 52727, Korea; 9854008@naver.com; 9Electronics and Telecommunications Research Institute, Daejeon 34129, Korea; hrlee@etri.re.kr

**Keywords:** lipocalin-2, amyloid-beta, neuroinflammation, iron accumulation, oxidative stress, blood–brain barrier leakage, Alzheimer’s disease

## Abstract

Lipocalin-2 (LCN2) is an inflammatory protein with diverse functions in the brain. Although many studies have investigated the mechanism of LCN2 in brain injuries, the effect of LCN2 on amyloid-toxicity-related memory deficits in a mouse model of Alzheimer’s disease (AD) has been less studied. We investigated the role of LCN2 in human AD patients using a mouse model of AD. We created an AD mouse model by injecting amyloid-beta oligomer (AβO) into the hippocampus. In this model, animals exhibited impaired learning and memory. We found LCN2 upregulation in the human brain frontal lobe, as well as a positive correlation between white matter ischemic changes and serum LCN2. We also found increased astrocytic LCN2, microglia activation, iron accumulation, and blood–brain barrier disruption in AβO-treated hippocampi. These findings suggest that LCN2 is involved in a variety of amyloid toxicity mechanisms, especially neuroinflammation and oxidative stress.

## 1. Introduction

The number of people with dementia is rapidly increasing, which is becoming a worldwide socioeconomic problem. Although Alzheimer’s disease (AD) is responsible for a significant fraction of all dementia cases, no disease-modifying treatment has yet been developed [1]. Understanding the many pathways involved in AD pathophysiology is critical to develop effective treatments, especially at early disease stages [2]. The abundant accumulation of amyloid-beta (Aβ) peptides and hyperphosphorylated tau protein in the brain are two major pathologies of AD [2]. In AD, amyloid precursor proteins are improperly cleaved by beta-secretase to form Aβ42, which form oligomers, which further assemble into fibrils and senile plaques [2]. The Aβ oligomer (AβO) is the most toxic species of the Aβ family and plays a role in disease progression by damaging neurons, glia, and blood vessels. However, the process by which AβO interacts with neurons, glia, and the vasculature is extremely complex and remains largely unclear [3]. 

Lipocalin-2 (LCN2), also known as neutrophil gelatinase-associated lipocalin, is an acute-phase protein that protects the body against infection by sequestering iron [4,5]. Recent research on the influence of LCN2 on memory impairment [4,5,6,7,8] found that LCN2 is engaged in cellular mechanisms of Aβ42-induced neuroinflammation and the proapoptotic signaling pathway, according to in vitro studies [3,4,9]. In the J20 AD mouse model, LCN2 affected iron dysregulation, but not cognition and plaque loading [8]. Using in vivo models of chronic metabolic conditions, such as obesity or cancer, LCN2 was associated with hippocampal neurotoxicity and cognitive impairment via neuroinflammation, oxidative stress, and blood–brain barrier (BBB) leakage [10,11]. In contrast to studies showing the detrimental effects of LCN2, some studies have reported that LCN2 maintains normal memory function and protects the brain from inflammation [12,13]. As a result, LCN2 has diverse functions as a neuroprotective or neurotoxic agent. In human studies, LCN2 expression was enhanced in post mortem AD brains, particularly in the hippocampus [3,14]. However, because LCN2 is modified by age, sex, obesity, and numerous metabolic variables, the results of serum or cerebrospinal fluid (CSF) LCN2 are ambiguous for memory impairment [4,6,7,15]. Despite several studies on the effects of LCN2 on memory impairment in various brain injuries, the response of LCN2 to AβO has been less studied in vivo models. The objectives of the present study were to determine the mechanism of LCN2 effects on AβO using an in vivo mouse model and to investigate the correlation of serum LCN2 with white matter ischemic changes in AD patients. The results of this investigation revealed that LCN2 is involved in neuroinflammation, iron-related oxidative stress, and BBB dysfunction in AD. 

## 2. Materials and Methods

### 2.1. Human Samples 

Participants were registered through the memory clinic at Gyeongsang National University Hospital (GNUH) from January 2018 to May 2021, and all registered participants submitted blood and clinical information to the GNUH Biobank. A total of 90 participants were examined; we excluded patients with diseases affecting cognition such as malignancy, alcohol use disorder, major depression, end-stage organ failure, previous stroke, and severe small vessel disease. Finally, this study included 17 healthy subjects and 37 patients with probable AD dementia. Subject demographics, brain magnetic resonance imaging (MRI), and serum LCN2 levels were all examined. All participants were required to sign a written informed consent form to participate in the study. This study was approved by the GNUH Institutional Review Board (IRB No. 2021-05-011). General cognition was assessed using the Korean version of the Mini-Mental State Examination (K-MMSE), clinical dementia rating, and global deterioration scale (GDS). Alzheimer’s disease dementia was diagnosed using the NINCDS-ARDRA criteria [16]. All subjects underwent a brain MRI using 3.0 Tesla units (Signa HDxt 1.5T; GE Healthcare, Milwaukee, WI, USA). The MRI protocol consisted of axial T1 and T fluid attenuated inversion recovery (FLAIR), coronal T1, and SWI. White matter changes were evaluated by one neurologist and one radiologist using a rating scale for age-related white matter changes (ARWMC) on the T2 axial and T2 FLAIR image [17]. The ARWMC scale divides the brain into five regions: frontal, parieto-occipital, temporal, basal ganglia, and infratentorial regions. On the T2 and FLAIR images, white matter alterations were classified as signals with intensity greater than 5 mm. The scales were scored from 0 to 3 depending on the extent of white matter changes, and total ARWMC was defined as the sum of each regional score [17].

### 2.2. Human Brain Autopsy 

Paraffin-embedded human frontal cortex sections of three AD (age 70–75, two apolipoprotein E (APOE)-e3 and one e4 homozygotes) and three non-disease (age 64–79, APOE e3 homozygotes), post mortem age-matched male individuals were received from the Victorian Brain Bank. 

### 2.3. Animals 

Male C57BL/6J mice were purchased from Central Lab Animal (Seoul, South Korea). These mice were maintained in the animal facility at Gyeongsang National University (GNU) before being used in the experiments. The Animal Care Committee for Animal Research at GNU approved the study protocol (GNU-160530-M0025). Mice were housed in a 12 h light/12 h dark cycle with free access to diet and water.

### 2.4. Oligomeric Amyloid-Beta Preparation and Infusion

AβO were prepared using the method of Mairet-Coello et al. [18]. Toxic Aβ aggregates were prepared and characterized for structural and functional AD research studies. Briefly, Aβ1-42 peptides (100 mΜ, Sigma-Aldrich, St. Louis, MO, USA) were dissolved in 100% dimethyl sulfoxide (DMSO) and ultrapure water was also added. A 2 M Tris–base solution (pH 7.6) was then added, and the mixture was incubated for 1–5 min at room temperature. Mice underwent stereotaxic surgery under deep anesthesia with Zoletil (40 mg/kg, Virbac Laboratories, Carros, France) and Rompun (5 mg/kg, Bayer Korea, Seoul, Korea). At 28 weeks of age, mice received one injection of AβO (1 mg/mL) on one side into the hippocampal CA1 region using a 10 µL Hamilton syringe and a fully motorized stereotaxic instrument (Neurostar, Animalab, Poland). The stereotaxic coordinates for the injection are referenced in −1.70 mm from the bregma according to the mouse brain atlas by *Paxinos and Franklin’s the Mouse Brain* (AP = −2.0 mm; ML = ±1.8 mm; DV = −1.5 mm). A 2 μL solution of Aβ (*n* = 15) or sterile saline (CTL, *n* = 15) containing DMSO (used as vehicle) was injected over 10 min. 

### 2.5. Morris Water Maze

The Morris water maze (MWM) is a classic test to examine spatial learning and memory. Six weeks after AβO injection, MWM testing was performed as previously described [19]. The test requires mice to find a hidden platform placed 1 cm below the milky opaque water (25 ± 0.5 °C) surface in one quadrant of a swimming pool (radius = 50 cm) that was divided into four quadrants. All mice performed four trials per day for four consecutive days, and the platform was removed on the day of testing. The escape latency and swimming distance needed to find the platform was recorded by a video-tracking program (Noldus EthoVision XT7; Noldus Information Technology, Wageningen, The Netherlands).

### 2.6. Enzyme-Linked Immunosorbent Assay (ELISA)

Human serum was isolated from whole blood via centrifugation and stored at −80 °C. Serum LCN2 was measured using a human LCN2 ELISA kit (R&D Systems, Minneapolis, MN, USA). 

### 2.7. Tissue Collection and Sample Preparation

For tissue analysis, mice (*n* = 4–5 per group) were intraperitoneally injected with Zoletil (40 mg/kg, Virbac Laboratories) and Rompun (5 mg/kg, Bayer Korea) and perfused with 4% paraformaldehyde in 0.1 M PBS. At 6 h after fixation, the brains were sequentially immersed in 15% and 30% sucrose at 4 °C until they were completely submerged. The brains were sliced into 30 µm coronal sections. 

### 2.8. Immunohistochemistry Analysis

Free-floating brain sections were incubated with primary antibodies described in Appendix A, followed by secondary biotinylated antibodies (Vector Laboratories, Burlingame, CA, USA). After washing, sections were incubated in an avidin–biotin–peroxidase complex solution (Vector Laboratories) and developed with diaminobenzidine (Sigma-Aldrich). Sections were dehydrated through graded alcohols, cleared in xylene, and mounted with Permount (Sigma-Aldrich). Images of the stained sections were captured using a BX51 light microscope (Olympus, Tokyo, Japan). 

### 2.9. Analysis of Microglia Morphology 

FIJI software and the Sholl analysis plugin were used for manual morphological analysis [20]. Images were uploaded into 16 bit format and slices containing ionized calcium binding adaptor molecule-1 (Iba-1)-positive cells were identified. A binary mask was created by thresholding the maximum intensity projections of the cells. The black area was measured, and the circularity index (CI) was computed (4π[area]/[perimeter]^2^). Sholl analysis was performed using the greatest radius of the cell soma and the radius exceeding the cell’s longest branch, as well as a manual count of primary branches. Furthermore, the ramification index (number of end branches/number of primary branches) was obtained for each cell using default settings of the FIJI and Sholl analysis plugin (Radius step size = automatic (“0.00”), Enclosing radius cutoff = 1 intersection, Sholl Method = linear). A minimum of 15 cells per imaging region was analyzed for each mouse.

### 2.10. Immunofluorescence

Free-floating brain sections were blocked for 1 h at room temperature with the corresponding blocking solution and incubated overnight at 4 °C with the corresponding primary antibodies listed in Appendix A. The next day, they were washed three times in 0.1 M PBS and incubated for 1 h in darkness at room temperature along with Alexa Fluor 488 and 594-conjugated secondary antibodies (Invitrogen Life Technologies, Carlsbad, CA, USA). The sections were counterstained with 4′,6-diamidino-2-phenylindole (DAPI, 1: 10,000, Invitrogen) and then washed in PBS and coverslipped using a mounting solution (Invitrogen). Images were acquired using a BX51-DSU microscope (Olympus).

### 2.11. Western Blotting

Western blot analysis was performed using standard procedures. Hippocampal tissues were collected from mice (*n* = 8–10 mice per group) and homogenized in lysis buffer (Thermo Scientific, Waltham, MA, USA). Proteins were immunoblotted with the corresponding primary antibodies, as shown in Appendix A. Anti-β-actin antibody served as a loading control to normalize the signal intensity of each target protein. The density of target bands was measured using the Multi-Gauge V 3.0 image analysis program (Fujifilm, Tokyo, Japan).

### 2.12. Statistical Analysis

IBM SPSS Statistics 21.0 (SPSS Inc., Chicago, IL, USA) was used in the human study. Frequency and percentage were used as categorical variables, whereas means and standard deviations were used as continuous variables. The Pearson χ2-test or Fisher’s exact test were used to evaluate categorical variables, and the independent t-test was used to analyze continuous variables. Linear regression analysis was used to account for the influence of age, sex, and education. The Pearson correlation coefficient was used to examine the relationship between LCN2 and age or ARWMC. In animal studies, group differences were determined using an unpaired t-test using PRISM 7.0 (GraphPad Software Inc., San Diego, CA, USA). Results were expressed as the means ± standard errors of the mean (SEM). *p* values < 0.05 were considered statistically significant. 

## 3. Results

### 3.1. Serum LCN2, White Matter Ischemic Changes on Brain MRI, and LCN2 Expression in Human Frontal Cortex Are All Increased in AD Patients

We examined general cognition, serum LCN2, and white matter ischemia alterations in 37 AD patients and 17 healthy controls (Table 1). The AD group was older than the normal control group (*p* < 0.05) and had more frequent hypertension and APOE e4 allele than the controls (*p* > 0.05). The AD patients showed lower folate levels and higher homocysteine levels than the controls (*p* < 0.05). Serum LCN2 was higher in the AD patients than in the controls (Table 1). White matter ischemic changes were more apparent in the AD patients (Table 1) and were mainly confined to the cerebral white matter in this group, not the basal ganglia or infratentorial regions (Figure 1A). After adjusting for age, sex, and education, however, there was no statistically significant difference in serum LCN2 or white matter ischemic changes between the two groups. In AD patients, white matter ischemic changes positively correlated with serum LCN2 (Pearson correlation coefficient = 0.336, *p* = 0.045) (Figure 1B) and homocysteine level (Pearson correlation coefficient = 0.423, *p* = 0.031). Immunofluorescence reveled that LCN2-positive glial fibrillary acidic protein (GFAP)-labeled astrocytes were increased simultaneously in the Aβ-positive amyloid plaques in the human frontal cortex of AD patients (Figure 1C). These findings indicate that astrocytic LCN2 is closely involved in the amyloid pathology. 

### 3.2. AβO-Injected Mice Have Memory Impairment

We stereotactically injected AβO into the hippocampus of 28-week-old mice and completed the MWM test at 34 weeks. In the MWM test, AD mice had impaired spatial learning and memory compared to wild-type animals and their learning slope was reduced (Figure 2A). The swim speed did not vary significantly between the two groups, indicating that the animals’ locomotor function was unaffected in the memory test (Figure 2B). The time spent in a target quadrant and the number of platform crossings were decreased (Figure 2C–E). The increased phosphorylation of tau (p-tau) as a marker of neurodegeneration was seen in the AD mice (Figure 2F), but amyloid plaques were not seen in the hippocampus (data not shown). These findings suggest that the AD mouse model is clinically and pathologically appropriate for investigating the role of LCN2 in AβO toxicity-induced memory impairment.

### 3.3. Activated Microglia, Astrocyte, and Pro-Inflammatory Cytokines Are Increased in the Hippocampus of AD Mouse Model

Because a sustained immunological response is a key factor in the progression of AD [21], we analyzed changes in microglia, astrocytes, and pro-inflammatory cytokines involved in the immune response. Microglial density, measured by Iba-1, was increased in the hippocampus of AD mice (Figure 2G). Microglia at rest had a small soma, fine and lengthy processes, and a lot of ramification (Figure 2H) [22]. Microglia in AD mice had shorter processes and a lower ramification index than the controls (Figure 2I,J), indicating microglial activation. Glial fibrillary acidic protein (GFAP) expression, which indicates astrogliosis, was also elevated in AD mice (Figure 3C,D). We found increased levels of IL-6, TNF-α, high mobility group box 1 (HMGB1), receptor for advanced glycation end products (RAGE), Toll-like receptor 4 (TLR4), and nuclear factor-kB (NF-kB) p65 in AD mice (Figure 3A,B). Another potential cause of neuroinflammation was increased Ly6G-positive neutrophils in the hippocampus of AD mice (Figure 3D). Neutrophils typically remain in blood vessels under normal conditions, but they migrated out of the astrocytes in the AD mice (Figure 3), suggesting a change in the BBB permeability.

### 3.4. The AD Mouse Model Exhibits Blood–Brain Barrier Leakage

BBB dysfunction occurs in the early stage of AD [23,24,25]. In addition to the leakage of neutrophils, we evaluated changes in the factors associated with BBB morphology and permeability. In Western blot and immunohistochemistry analyses, exposure to AβO compromised zonula occludens-1 (ZO-1), resulting in molecular alterations at the BBB tight junction (Figure 4). Increased albumin levels were reported in the AD mouse model, indicating that the BBB had been disrupted on a wide scale (Figure 4A,B). Elevated endothelial nitric oxide synthase (eNOS) and vascular adhesion protein 1 (VCAM-1) protein levels (Figure 4A,B) are consistent with the promotion of greater BBB permeability.

### 3.5. Increased Expression of LCN2, MMP-9, and STAT-3 in the AD Mouse Model

Matrix-metalloproteinase 9 (MMP9) is known to increase BBB permeability and activate neuroinflammation [26] and MMP9 was increased by intracerebral injection of Aβ in the mouse hippocampus [27]. In a previous study, we showed that MMP9 and LCN2 expression increased simultaneously in hippocampal neurons of ob/ob mice, and that both were located in close proximity [11]. Thus, we first determined hippocampal LCN2 protein. LCN2 expression was significantly in AβO-treated mice compared to controls (Figure 5A,B). MMP9 did also increase in the hippocampus of AD mice, as did phosphorylated signal transducer and activator of transcription 3 (STAT3) (Figure 5A,B). LCN2 expression was observed in GFAP-labeled astrocytes (Figure 5C). LCN2 from astrocytes is known to activate STAT3, and activated STAT3 promotes the production of pro-inflammatory cytokines, including MMP9, TNF-α, and IL-6 [28,29].

### 3.6. Increased Iron Dysregulation and Oxidative Stress in AD Mouse Model

LCN2 is an iron-modulating protein in brain [30]. Abnormal iron accumulation is associated with oxidative stress, amyloid plaque formation, apoptosis, and neuroinflammation [31]. Heme oxygenase 1 (HO-1) is an inducible enzyme that produces free iron and is involved in heme degradation and metabolism [32]. Perls’ staining for iron detection showed markedly increased iron accumulation in AD mice compared to control mice (Figure 6A). Both ferritin, a storage iron, and ceruloplasmin, which acts on iron oxidation and efflux, increased (Figure 6B). HO-1 was increased in AD mice (Figure 6B), indicating an increase in free iron and iron-related oxidative stress [33]. 

## 4. Discussion

Many recent studies have found that LCN2 affects cellular mechanisms and cognitive function in neurodegenerative disorders [4,5,9,10]. LCN2 had varying effects in the brain depending on the type of injury or the amount of time that the brain was exposed to LCN2 [8,10,11,12,13]. Acute and short-term LCN2 exposure has a neuroprotective effect, while chronic LCN2 exposure has a neurotoxic effect [10,13]. 

Using the AβO-injected AD mouse model, we demonstrated cognitive decline and neurodegeneration with increased p-tau, but no amyloid fibrils or plaques in the hippocampus. Amyloid peptides in the brains of humans and mice are constantly cleared through phagocytosis, proteolysis, and efflux to the peripheral circulation [34,35]. Because our study used just a single injection of AβO, we believe that amyloid peptides were eliminated via a clearance process after injection. However, because a considerable amount of AβO was injected directly into the hippocampus, amyloid toxicity would have progressed through the immune system. This could explain why AD progression continues after amyloid removal from the brain using monoclonal antibodies and may inform the development of AD drugs [1]. 

The immune response has emerged as a key player in the initiation and progression of AD, causing researchers to become increasingly interested in astrocytes and microglia that play a role in innate immunity [21,36]. In our study, AβO injection stimulated astrocytes and microglia. Astrocyte is the most abundant glial cell in the brain. GFAP is the main constituent of the astrocytic cytoskeleton, overexpressed during reactive astrogliosis in AD [37]. As shown in Figure 3D, morphological hypertrophy and increased GFAP contents are regarded as reflections of a detrimental astrocyte phenotype. The microglia is known to monitor the microenvironment in its resting state and to be activated by pathogen invasion or injury [34]. Activated microglia removes pathogens, such as Aβ peptides, by phagocytosis, using scavenger receptors such as TLR4 [34]. Microglia, on the other hand, promotes inflammation through the NF-kBp65 pathway, which plays a crucial role in AD pathogenesis when there are iron-related oxidative stress or chemotactic factors [36]. Thus, we focused on LCN2 as a factor that keeps microglia active. 

In this study, LCN2 was increased in the serum and brain tissue of AD patients, as well as in the astrocytes of an AD mouse model. In previous in vitro research, Aβ produced LCN2, which then triggered the TNF-α-mediated neurotoxic pathway [4,5,12]. Previous studies on the link between LPS and LCN2 relied mostly on LPS-induced neuroinflammation [10,13,38]. These findings implicate LCN2 as a proinflammatory mediator of memory deficits. So, a future study should investigate LCN2 expression and cognitive impairment after intrahippocampal injection of LPS in order to clarify the role of LCN2. In this study, we found higher levels of HMGB1, TLR4, RAGE, NF-kBp65, and TNF-α in the hippocampus in the AD mouse model. HMGB1 as well as TNF-α have roles in initiating inflammation by binding to TLR4 and RAGE and activating NF-kBp65, which induces LCN2 gene expression [39,40,41]. Thus, injected AβO is initially recognized by microglia that then produce inflammatory cytokines such as TNF-α and HMGB1 that boost LCN2 expression by activating NF-kBp65. LCN2 acts as an autocrine or paracrine activator of microglia and astrocytes.

LCN2 is implicated in the internalization and accumulation of iron from in vitro and in vivo studies [8,11,30]. Iron levels in the brain rise with age and are linked to the development of neurodegenerative disorders such as AD, Parkinson’s disease, and others [42]. In energy metabolism, iron is a component of mitochondrial enzymes that produce ATP and use oxygen. The Fenton reaction produces hydroxyl radicals and reactive oxygen species (ROS), which harm lipids and DNA as iron levels rise. Inflammatory circumstances enhance LCN2 expression, and high LCN2 increases iron overloading [42]. In this study, serum homocysteine levels increased while folate levels decreased in the AD patients. Homocysteine is involved in methylation, glutathione biosynthesis, and mitochondrial oxidative stress. Glutathione synthesis is inhibited by hyperhomocysteinemia. Reduced glutathione promotes iron liberation from ferritin, resulting in more free iron and iron-dependent ROS [43,44]. We detected increased LCN2, iron buildup and iron metabolism-related enzymes in this AβO-injected mouse model, similar findings to the prior J20 Alzheimer mouse study [8]. However, in the J20 AD mouse study, it was reported that LCN2 had no effect on cognition and glial activation. The pathology of the two animal models differs slightly from that of human AD. This AD mouse model showed p-tau expression but no amyloid plaques, whereas the J20 mouse model had amyloid plaques but no tau-tangles. A study using the J20 mouse model showed different results from this study in cognitive impairment and glial activation because a significant amount of amyloid plaque overwhelmed the effect of LCN2. However, both models showed increased LCN2 and iron accumulation, suggesting that this pathway may be a primary mechanism of memory impairment. Free iron overloading is induced by upregulated LCN2 and HO-1 in microglia, producing reactive oxygen species and oxidative stress [30,31,33,36] that lead to the pathologic activation of microglia [36]. A neuroimaging study showed that iron-containing microglia were found in the hippocampus of AD patients [45].

Serum and CSF LCN2 levels were higher in patients with vascular dementia [46,47] and increased CSF LCN2 positively correlated with age-related white matter alterations [46]. Although white matter changes are widespread in the brains of AD patients [48], the link between serum LCN2 and white matter changes in clinically probable AD patients has not been studied. Here, we found a link between serum LCN2 and age-related white matter ischemic changes in AD patients. White matter ischemic changes have been associated with BBB disruption [32]. In a study using a CSF biomarker and dynamic contrast-enhanced MRI, brain capillary damage and BBB disruption developed in the early stage of cognitive dysfunction, irrespective of Alzheimer’s two pathologic hallmarks [49,50]. The link between LCN2 and BBB impairment has been established in several reports [46,47,51,52,53]. After focal cerebral ischemia, Du et al. found that LCN2 restored endothelial permeability and ZO-1 expression [50]. In other studies, LCN2 reduced endothelium tight junction protein and worsened BBB permeability [46,47,52,53]. We found decreased ZO-1 in the endothelium and increased albumin in the hippocampus of this AD model, suggesting BBB leakage. We also found increased peripheral neutrophils in the hippocampus, which is mediated by VCAM-1 and eNOS. MMP9 is associated with BBB permeability, and the inhibition of MMP9 reduces Aβ-induced cognitive impairment [26,27]. Increased LCN2 and HMGB1 levels are associated with MMP9 expression via STAT3 and NF-kB activation [28,29,54]. Increased LCN2 binds to MMP9 and suppresses its degradation, allowing MMP9 to function for longer periods of time, thereby disrupting angiogenesis and the BBB [11,55]. BBB disruption exacerbates neuroinflammation and activates the glial system even more.

This study has several limitations. One is that the human trial had a small sample size and did not quantify CSF LCN2. The other is that the impact of LCN2 was not validated in the animal study using the LCN2 null or an in vitro investigation. Despite these limitations, this study indicates that LCN2 and iron are involved in sustained glial activation, and that activated glial cells play a role in AβO-induced memory impairment via neuroinflammation, iron-related oxidative stress, and BBB disruption.

## 5. Conclusions

The results of this study suggest that AβO activates microglia and causes astrocytic LCN2-mediated neuroinflammation, which includes increased pro-inflammatory cytokine release, iron-specific oxidative stress, and BBB leakage. Therefore, these findings indicate that although LCN2 is not a specific response to AβO, it plays a key role in the progression of AD pathology caused by AβO.

## Figures and Tables

**Figure 1 antioxidants-10-01657-f001:**
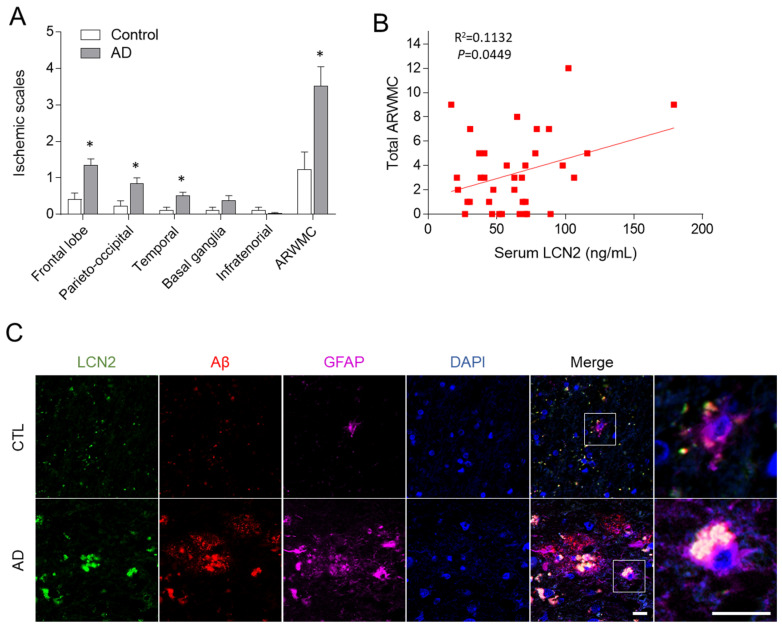
Human LCN2 expression and white matter ischemic changes in AD patients. (**A**) Assessment of ischemic rating scales in control and AD patients. (**B**) Correlation between total ARWMC and serum LCN2 levels in AD patients. (**C**) Representative immunofluorescence images (×200) showing LCN2 (green), Aβ (red), and GFAP (purple) staining in frontal cortex sections from an AD patient. DAPI (blue) was used to stain nuclei. Scale bar = 10 µm. Data are shown as mean ± SEM. * *p* < 0.05 versus control (CTL).

**Figure 2 antioxidants-10-01657-f002:**
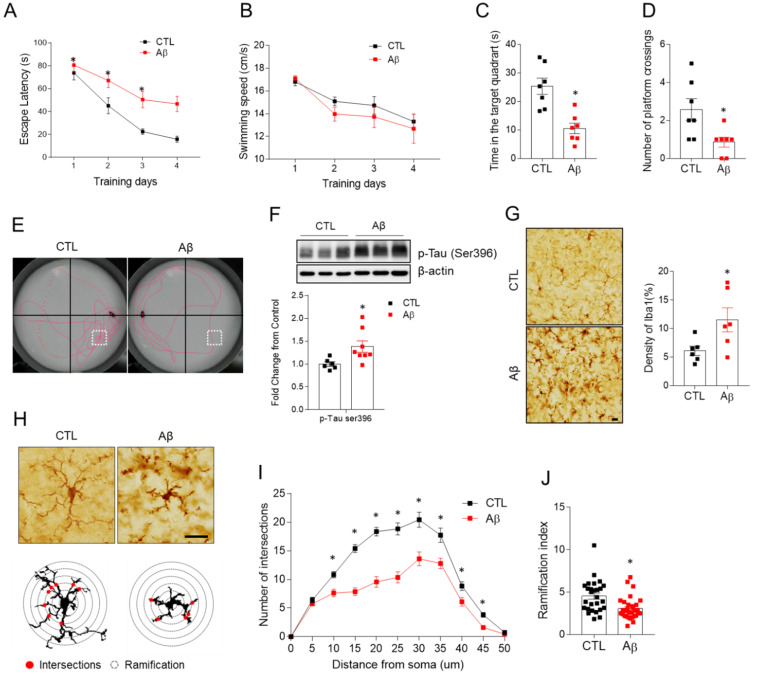
Effects of AβO toxicity on memory deficits, hippocampal tau phosphorylation and microglia activation. The Morris water maze test in control and Aβ-treated mice at 34 weeks. (**A**) Escape latency and (**B**) average time spent in swimming speed over 4 days. (**C**) Time spent in the target quadrant, and (**D**) number of crossings over the platform area for the three time slices analyzed. (**E**) Representative swimming paths during the probe trail. (**F**) Western blot and quantified hippocampal phosphorylated tau (s396) expression. To normalize total protein level, β-actin was used as a loading control. (**G**) Representative images of Iba-1 immunohistochemistry and quantification of relative optical density (ROD) measurements (%) in hippocampal CA1 regions. Bar = 10 µm (**H**). The schemes graphically illustrate Sholl analysis of microglia morphology detects. (Red circles indicate Sholl intersections. Circle lines indicate Sholl sphere radius). (**I**) Average number of intersections at specified distances from the soma in microglia, and (**J**) Shoenen ramification index; 27 to 31 cells per region of *n* = 4 mice. Data are mean ± SEM. * *p* < 0.05 for control compared with Aβ-treated mice.

**Figure 3 antioxidants-10-01657-f003:**
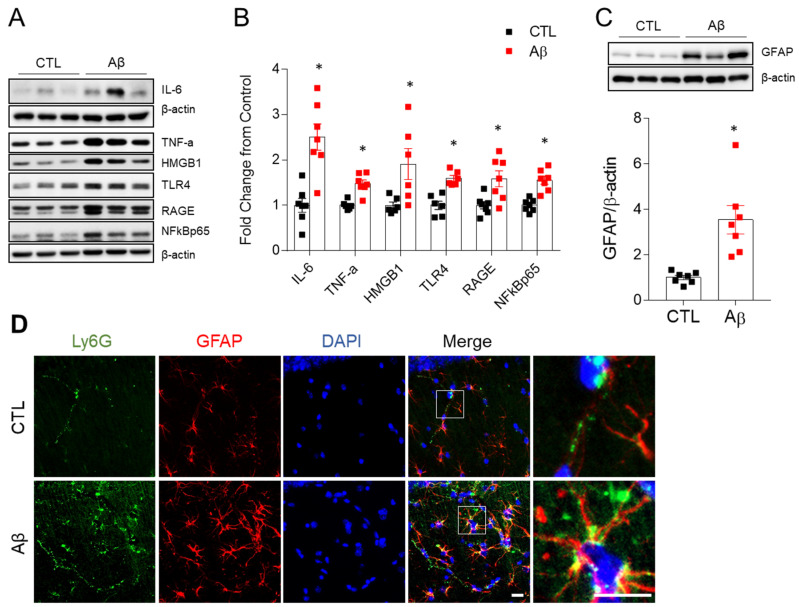
Effects of AβO toxicity on neuroinflammation in the hippocampus. (**A**) Western blot analysis of IL-6, TNF-α, HMGB1, TLR4, RAGE, and NF-κBp65 in the hippocampus. (**B**) Quantitation of Western blot analysis in (**A**). To normalize total protein level, β-actin was used as a loading control. (**C**) Western blot and quantified hippocampal GFAP expression. To normalize total protein level, β-actin was used. The data are presented as mean ± SEM. * *p* < 0.05 for control compared with Aβ-treated mice. (**D**) Representative immunofluorescence images (×200) of hippocampal CA1 regions showing Ly6G (green) and GFAP (red) staining. DAPI (blue) was used to stain nuclei. Scale bar = 10 µm.

**Figure 4 antioxidants-10-01657-f004:**
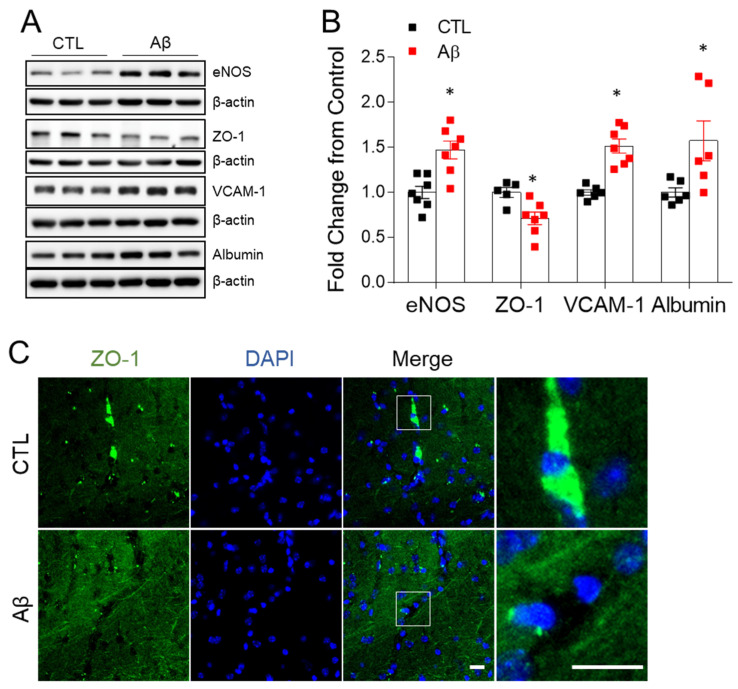
Effects of AβO toxicity on BBB leakage in the hippocampus. (**A**) Western blot analysis of eNOS, ZO-1, VCAM-1, and albumin in the hippocampus. (**B**) Quantitation of Western blot analysis in (**A**). To normalize total protein level, β-actin was used as a loading control. The data are presented as mean ± SEM. * *p* < 0.05 for control compared with AβO-treated mice. (**C**) Representative immunofluorescence images (×200) of hippocampal CA1 regions showing ZO-1 (green). DAPI (blue) was used to stain nuclei. Scale bar = 10 µm.

**Figure 5 antioxidants-10-01657-f005:**
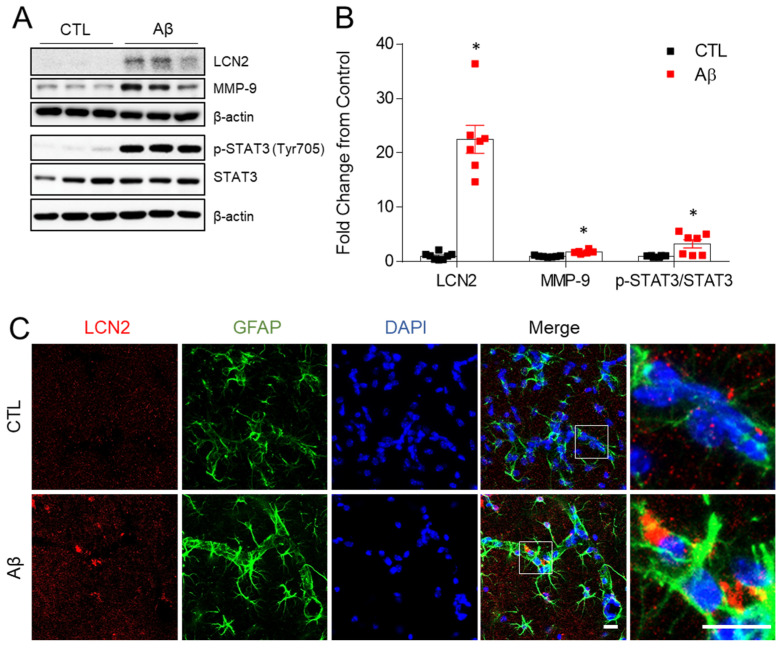
Effects of AβO toxicity on LCN2, MMP9, and STAT3 expression in the hippocampus. (**A**) Western blot analysis of LCN2, MMP-9, p-STAT3 (Tyr705), and STAT3. (**B**) Quantitation of Western blot analysis in (**A**). To normalize total protein level, β-actin was used as a loading control. The data are presented as mean ± SEM. * *p*< 0.05 for control compared with AβO-treated mice. (**C**) Representative immunofluorescence images (×200) of hippocampal CA1 regions showing LCN2 (red) and GFAP (green). DAPI (blue) was used to stain nuclei. Scale bar = 10 µm.

**Figure 6 antioxidants-10-01657-f006:**
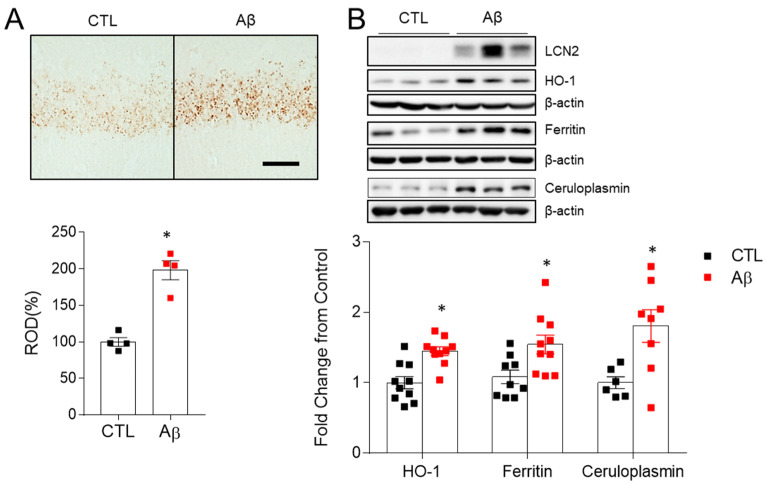
Effects of AβO toxicity on iron accumulation and oxidative stress in the hippocampus. (**A**) Histological staining (×100) for iron with DAB-enhanced Perls’ staining in the hippocampal CA1 regions and relative optical density (ROD) measurements (%). Scale bar = 25 µm. (**B**) Western blots and quantified hippocampal HO-1, Ferritin, and Ceruloplasmin expressions. β-actin was used as a loading control. As shown in Figure 5A, AβO-induced LCN2 is also enhanced in same Western blot used in HO-1. The data are presented as mean ± SEM. * *p* < 0.05 for control compared with AβO-treated mice.

**Table 1 antioxidants-10-01657-t001:** Characteristics of control and Alzheimer’s disease patients.

	Controls (*n* = 17)	Alzheimer’s Disease (*n* = 37)	*p*1-Value	*p*2-Value
Age (years)	71.8 ± 8.1	78.1 ± 6.4	0.003	
Sex (female:male)	16:1	25:12	0.043	
Education (years)	8.89 ± 4.27	6.69 ± 5.01	0.255	
HbA1C (%)	5.6 ± 0.4	5.6 ± 0.3	0.525	
Hypertension (%)	4 (23.5%)	16 (43.2%)	0.229	
K-MMSE	27.6 ± 2.7	17.7 ± 4.7	<0.001	
GDS	2.1 ± 0.7	4.1 ± 0.8	<0.001	
ApoE e4 genotype (%)	2/13 (15.4%)	11/24 (45.8%)	0.083	
Folate (ng/mL)	11.9 ± 5.0	7.1 ± 4.5	0.006	0.064
Vitamin B12 (pg/mL)	1133.8 ± 538.2	876.9 ± 466.6	0.149	0.134
Homocysteine (µmol/L)	7.7 ± 1.1	13.4 ± 3.9	<0.001	<0.001
LCN2 (ng/mL)	42.1 ± 22.7	63.8 ± 31.8	0.018	0.147
BDNF (pg/mL)	11.3 ± 3.7	12.5 ± 5.0	0.36	0.116
ARWMC (score)	1.4 ± 2.0	3.5 ± 3.2	0.007	0.186

HbA1c, hemoglobin A1c; K-MMSE, Korean version of the Mini-Mental State Examination; GDS, global deterioration scale; LCN2, lipocalin-2; BDNF, brain-derived neurotrophic factor; ARWMC, age-related white matter changes. *p*1 values were calculated by independent *t*-test or Pearson χ2-test. *p*2 values were calculated by linear regression model adjusted by age, sex, and education.

## Data Availability

The data presented in this study are available in this manuscript.

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
