# Peer review of "Role of Lipocalin-2 in Amyloid-Beta Oligomer-Induced Mouse Model of Alzheimer’s Disease"

_antioxidants, 2021, doi:10.3390/antiox10111657_

Round 1

Reviewer 1 Report

The presented work obtains important conclusions regarding the relationship between LCN2 and neuroinflammation. However, some issues must be clarified prior to publication.

Although the role of LCN2 in Aβ1-42 toxicity is clear and proven, it cannot be stated that it is due solely to its effect on amyloid peptide and AD. The effect of LCN2 may be due to its role on neuroinflammation in general. In fact, this protein is also important in other diseases in which amyloid peptide does not participate and present neuroinflammation. Some examples are cerebral ischemic stroke, steatohepatitis, iron-mediated oxidative stress, amyotrophic lateral sclerosis, diabetic encephalopathy, etc. For these reasons, the conclusions and the general approach to the work must be modified in this sense. Although the mouse model has been injected with Aβ1-42 and has cognitive deficit, the effect of LCN2 may still be mediated by the neuroinflammation. The direct relationship between LCN2, amyloid peptide and AD is not demonstrated.

Methodologial issues

The level of LCN2 in serum is not determined for healthy controls. What are the levels in the general population?.

The J20 mouse could also have been used in the study. As indicated in the study, this model and the J20 show important differences. Furthermore, in Dekens et al, 2018, it is concluded that LCN2 has no effect on cognition or glial activation. It should also be discussed why different results are obtained in this work (plaque formation and phosphorylation of Tau). All these differences could have served to clarify the mechanisms if this mouse had been included in the study. Also a different toxic should have been used. The use of LPS as an inflammation inducer would clarify whether the effects are due to amyloid peptide or to neuroinflammation.

LCN2 levels are studied with different techniques depending on the tissue or cell type. In the case of serum, it is observed by ELISA in humans and by WB in mice. In tissue it is determined by IHC in humans  in mice, but not by WB. Authors should explain why different methods are used for each case.

Astrocyte activation, although smaller, is also observed in controls. the authors should discuss the cause.

Hypertension is mentioned as a clinical feature but does not appear in the table. The adjusted p-values should also be determined for sex (not paired) and the adjusted p-values should be shown in the table in both cases. The characteristics of the patients used for the post-mortem brain analysis should also be detailed.

"Thus, we first determined circulating LCN2 levels (Figure 5A) but LCN2 protein did not significantly differ between controls and AβO-treated mice however". In figure 5A it is shown as significant.

If DAPI is used to quantify the expression in IHC it should be indicated in the corresponding section of Methods.

The marker proteins in section 3.6 are actually specific for Fe stress, not general oxidative stress.

LCN2 expression should be included in all WBs as a control.

Minor issues:

For the sentence "We also discovered elevated LCN2 in astrocytes, activated astrocytes and micro-glia, pro-inflammatory cytokines and receptors, iron dysregulation, and blood-brain barrier break-down", , the meaning of the sentence is lost. In scientific English it is better to write short sentences without subordinate phrases.

This work should be cited and discussed DOI: 10.3389/fnagi.2021.663837

Reviewer 2 Report

Presented manuscript intends to analyze role of lipocalin 2 in the etiology of AD in the amyloid beta mice model as well as in the human brain samples  with verified AD. Paper has a reasonable structure, originality of the approach is clearly visible  and results seem sound.  However, reviewer still feels that characteristic of human controls and AD patients is not well balanced, such as 1. sex differences, 2.the genotype of APO E, as well as the 3. levels of iron and iron binding capacity. Additional data such as homocysteine level can also be relevant for etiology pattern.  Discussion should also include the prominent  role of lipocalin in the iron metabolism and metabolic aspects linked with one carbon- homocysteine and altered redox balance

Round 2

Reviewer 1 Report

The authors have fixed most of  he major issues raised by the reviewer, but not all of them:

Methodologial issues

  1. The level of LCN2 in serum is not determined for healthy controls. What are the levels in the general population?

→ The normative data of circulating LCN2 level has not yet been validated. In the previous studies, Serum LCN2 level of the normal control has been shown in case-control or cohort studies [1-5]. Maybe, because LCN2 levels are influenced by various factors such as age, sex, or comorbidities, the LCN2 level measured by ELISA differs from study to study. Also, a methodological issues such as batch to batch variation cannot be excluded. We revised table 1.

The data showed in the author's reply letter do not appear in table 1

  1. Also a different toxic should have been used. The use of LPS as an inflammation inducer would clarify whether the effects are due to amyloid peptide or to neuroinflammation.

→ Thanks for your important comments. Previous research on the link between LPS and LCN2 relied mostly on LPS-induced neuroinflammation [1-3]. These findings implicate LCN2 as a proinflammatory mediator of memory deficits. As the reviewer suggested, a future study should be need to look at LCN2 expression and cognitive impairment after intracerebral injection of LPS in order to clarify the mechanism.

It can be stated in this way in the discussion.

  1. Astrocyte activation, although smaller, is also observed in controls. the authors should discuss the cause.

→ Astrocyte is the most abundant glial cell in the brain. GFAP antibody is usually used as popular astrocyte marker. GFAP is the main constitute of the astrocytic cytoskeleton, overexpressed during reactive astrogliosis in AD. In our study, hippocampal LCN2 levels in AβO-treated mice were prominently increased compared to control mice. As shown in figure 3D, morphological hypertrophy and increased GFAP contents were generally regarded as reflections of a detrimental astrocyte phenotype. Thus, this is assumed to be responsible for the minor activation of astrocytes in the control group.

It can be stated in this way in the discussion.

  1. LCN2 expression should be included in all WBs as a control.

→ For example, we checked different MMP9, LCN2, and β–actin in same blots. To normalize total protein level, β-actin was used as loading control. So, we suggest that LCN2 expression does not need to be used as control for all WBs. Furthermore, we think that it is not necessary to include LCN2 in all WBs because LCN2 knockout mice were not used in this study.

I don't understand the explanation. What the reviewer proposes is to ensure the increase in LCN2 expression by WB, not to use it as a loading control.

Suggested text changes for "Conclusions"

In conclusion, tThe results of this study suggest that AβO activates microglia activation and causes astrocytic LCN2-mediated neuroinflammation, which includes increased pro-inflammatory cytokine release, iron-specific oxidative stress, and BBB leakage. So that/Therefore, these findings indicate that although LCN2 is not a specific response to AβO, LCN2 may play a key role in the progression of AD pathology caused by AβO.
